# Thymidine Analogue Mutations with M184V Significantly Decrease Phenotypic Susceptibility of HIV-1 Subtype C Reverse Transcriptase to Islatravir

**DOI:** 10.3390/v16121888

**Published:** 2024-12-06

**Authors:** Hyeonah Byun, Maria Antonia Papathanasopoulos, Kim Steegen, Adriaan Erasmus Basson

**Affiliations:** 1HIV Pathogenesis Research Unit, Faculty of Health Sciences, University of the Witwatersrand, Johannesburg 2193, South Africa; 1830847@students.wits.ac.za (H.B.); maria.papathanasopoulos@wits.ac.za (M.A.P.); 2National Priority Programme, National Health Laboratory Service, Johannesburg 2192, South Africa; kim.steegen@nhls.ac.za; 3Department of Molecular Medicine and Haematology, School of Pathology, Faculty of Health Sciences, University of the Witwatersrand, Johannesburg 2193, South Africa

**Keywords:** HIV, antiretroviral therapy (ART), antiviral drugs, NRTI, islatravir (ISL), phenotypic, drug resistance, subtype C

## Abstract

Islatravir (ISL) is the first-in-class nucleoside reverse transcriptase translocation inhibitor (NRTtI) with novel modes of action. Data on ISL resistance are currently limited, particularly to HIV-1 non-B subtypes. This study aimed to assess prevalent nucleos(t)ide reverse transcriptase inhibitor (NRTI)-resistant mutations in HIV-1 subtype C for their phenotypic resistance to ISL. Prevalent single and combinations of NRTI-resistant mutations were selected from a routine HIV-1 genotypic drug resistance testing database and introduced into HIV-1 subtype C-like pseudoviruses, which were then tested for ISL susceptibility. Single NRTI-resistant mutations were susceptible or showed only a low level of resistance to ISL. This included thymidine analogue mutations (TAMs, i.e., M41L, D67N, K70R, T215FY, and K219EQ) and non-TAMs (i.e., A62V, K65R, K70ET, L74IV, A114S, Y115F, and M184V). Combinations of M184V with one or more additional NRTI-resistant mutations generally displayed reduced ISL susceptibilities. This was more prominent for combinations that included M184V+TAMs, and particularly M184V+TAM-2 mutations. Combinations that included M184V+K65R did not impact significantly on ISL susceptibility. Our study suggests that ISL would be effective in treating people living with HIV (PLWH) failing tenofovir disoproxil fumarate (TDF)/lamivudine (3TC) or TDF/emtricitabine (FTC)-containing regimens, but would be less effective in PLH failing zidovudine (AZT) with 3TC or FTC-containing regimens.

## 1. Introduction

The human immunodeficiency virus (HIV) continues to be a significant threat to human health globally. HIV has infected an estimated 85.6 million individuals worldwide and the acquired immunodeficiency syndrome (AIDS) has claimed the lives of approximately 40.4 million people [1]. However, substantial advances have been made in the treatment of people living with HIV (PLWH). Combination antiretroviral therapy (cART) has proven to be effective in managing HIV infections to allow for a near-normal quality and span of life in PLWH [2,3]. Antiretroviral drugs (ARVs) cannot cure an HIV infection and cART is required for life to ensure sustained viral suppression. However, optimal cART adherence can be difficult to maintain in the long-term [4], with sub-optimal adherence driving the development of antiretroviral drug resistance [5]. Alternative treatment options and novel ARVs with efficacy against resistant viral variants are required that could address these shortfalls.

Islatravir (ISL, EFdA, or MK-8591) is a novel nucleoside reverse transcriptase translocation inhibitor (NRTtI) with a unique dual mechanism of action. Apart from acting as an atypical chain terminator, it also prevents HIV Reverse Transcriptase from translocating further on the viral RNA template [6]. ISL was shown to be well tolerated and effective in several clinical trials investigating dual-drug therapies [7]. Early studies on ISL showed an in vitro potency that was 10-fold higher than the approved NRTIs tenofovir alafenamide (TAF), zidovudine (AZT), and lamivudine (3TC) [8]. The common HIV-1 NRTI mutation M184V showed a nine-fold decrease in ISL susceptibility in HIV-1 subtype B NL4-3 [9]. However, the K65R, L74V, and Q151M mutations were more susceptible to ISL [8]. ISL was also shown to retain its potency against the multi-drug-resistant HIV-1 strain A62V/V75I/F77L/F116Y/Q151M [10], which is resistant to all current NRTIs except tenofovir [11].

Due to its long half-life, ISL is being investigated for use in extended dosing intervals, such as weekly dosing in combination with Lenacapavir [12]. However, dose-dependent decreases in the CD4 cell and total lymphocyte counts in ISL-containing regimens in some of these clinical trials [13,14] set the stage for the re-evaluation of ISL at lower doses. ISL has a high genetic barrier to the development of resistance and has shown to be effective against variants with major single nucleos(t)ide reverse transcriptase inhibitor (NRTI) drug-resistance mutations (i.e., M41L, K65R, L74I, and V90I) [15,16].

Since ISL is still in an early stage of clinical development, most data available on ISL drug resistance rely largely on observations in HIV-1 subtype B. However, the high genetic diversity among HIV-1 subtypes could have an impact on drug resistance selection pathways, and subsequently, the degree of ARV resistance imposed by resistance mutations [17]. HIV-1 subtype C predominates globally [18] and is the predominant circulating subtype among South Africa’s 7.7 million PLWH [19]. Currently, all recommended cART regimens in South Africa include at least two NRTIs, some of which (i.e., lamivudine) are often recycled between regimens [20]. To inform HIV clinicians and caregivers on the potential role of ISL in cART in the context of HIV-1 subtype C NRTI-experienced PLWH, this study evaluated the in vitro susceptibility of prevalent NRTI drug-resistance mutations to ISL.

## 2. Materials and Methods

### 2.1. Antiretroviral Drugs

Islatravir was purchased from MedChemExpress (Ipswich, MA, USA; Cat. No. HY-104012). The following reagent was obtained through BEI Resources, NIAID, NIH: Lamivudine (3TC), HRP-8146, contributed by the NIH HIV Reagent Program. ISL and 3TC were prepared in dimethyl sulfoxide (DMSO; Sigma-Aldrich, St. Louis, MO, USA; Cat. No. 472301) as 10 mM stocks and diluted into Dulbecco’s Modified Eagle Medium (DMEM; Thermo Fisher Scientific, Waltham, MA, USA; Cat. No. 11995065) to a working solution of 0.5 μM and 100 μM, respectively.

### 2.2. Vectors

The HIV-1 gag–pol expression vectors (i.e., p8.9MJ4, p8.9NSX+) were obtained from Didier Trono (École Polytechnique Fédérale de Lausanne, Lausanne, Switzerland) and Deenan Pillay (University College London, London, UK). The vectors have been modified by introducing a *PvuI* endonuclease restriction site at the start (nucleotides 11–16) of *reverse transcriptase*, and an *HpaI* endonuclease restriction site toward the end (nucleotides 1100–1106) of *reverse transcriptase*. The firefly luciferase transfer vector (i.e., pCSFLW) was obtained from Nigel Temperton (Medway School of Pharmacy, Gillingham, United Kingdom). The following reagent was obtained through the NIH HIV Reagent Program, Division of AIDS, NIAID, NIH: Plasmid pHEF Expressing Vesicular Stomatitis Virus (VSV-G), ARP-4693, contributed by Dr. Lung-Ji Chang.

### 2.3. Laboratory-Adapted Strains

The following laboratory-adapted HIV-1 strains were obtained from the National Institute for Communicable Diseases (NICD; Modderfontein, Johannesburg, South Africa) as frozen culture supernatants: HIV-1 subtype C—Du151, Du179, Du422, CM9; HIV-1 subtype B—SM1, SM2, DS9, LTNP5 (Appendix A).

### 2.4. HEK293T Cell Culture

The HEK293T cell line was maintained in complete DMEM (Thermo Fisher Scientific, Waltham, MA, USA; Cat. No. 11995065), containing 10% foetal bovine serum (Merck, Darmstadt, Germany; Cat. No. F0679), 25 mM D-glucose, 4 mM L-glutamine, 1 mM sodium pyruvate, and 0.25 mg/mL gentamicin (Thermo Fisher Scientific, Waltham, MA, USA; Cat. No. 15750060). The cells were cultured at 37 °C under 5% CO_2_ in a humidified incubator and passaged every two to three days.

### 2.5. Selection of NRTI Drug-Resistance Mutations

HIV-1 *reverse transcriptase* sequences (*n* = 7749) from routine genotypic drug resistance testing, performed between January 2016 and December 2020, were obtained from the National Health Laboratory Services (NHLS; Charlotte Maxeke Johannesburg Academic Hospital, Johannesburg, South Africa). The NHLS is a South African governmental institution that performs various serological and pathological tests on patients in the South African national health sector. Tests are requested by healthcare providers and performed with the consent of the patient. For this study, the use of anonymized HIV genotypic drug resistance data was approved by the University of the Witwatersrand’s Human Research Ethics Committee (Protocol M221081, ethics number R14/49). The twenty most prevalent NRTI drug resistance mutation combinations were identified and assessed for in vitro ISL susceptibility.

### 2.6. Preparation of Mutant Pseudoviruses in Reference Isolates

HIV-1-like replication defective pseudoviruses (PSVs) with the most prevalent combinations of NRTI mutations (*n* = 20) were prepared. In addition, PSVs containing the single mutations that constituted the combinations, as well as the A114S and A114S+M184V PSVs, were also generated. NRTI-drug-resistant mutations were introduced into the HIV-1 subtype C gag–pol expression vector (i.e., p8.9MJ4) through polymerase chain reaction (PCR)-based site-directed mutagenesis (SDM) using the Q5 Site-Directed Mutagenesis Kit (New England Biosciences, Ipswich, MA, USA; Cat. No. E0554), according to the manufacturer’s instructions. The mutagenesis primers were designed using NEBaseChanger™ v.1.3.3. (https://nebasechangerv1.neb.com/, accessed on 26 April 2022) (Appendix A). Multiple rounds of SDM were performed to incorporate more than one mutation into a single vector.

Chemically competent DH5α bacterial cells (New England Biosciences, USA; Cat. No. C2987H) were transformed with the SDM reactions through standard heat-shock. The transformed bacteria were plated on carbenicillin-containing (100 μg/mL; Thermo Fisher Scientific, Waltham, MA, USA; Cat. No. 10177012) Luria Bertani (LB; Thermo Fisher Scientific, Waltham, MA, USA; Cat. No. 22700025) agar plates and incubated for approximately 16 h at 37 °C. Carbenicillin-containing (100 μg/mL) LB broth (Thermo Fisher Scientific, Waltham, MA, USA; Cat. No. 12780052) was inoculated with bacterial colonies and incubated for approximately 16 h at 37 °C with agitation. Bacterial cells were pelleted through centrifugation at 4500 rpm for 10 min at room temperature. The bacterial pellets were used to extract plasmid DNA with the QIAprep Spin Miniprep Kit (QIAGEN, Hilden, Germany; Cat. No. 27106). The eluted plasmid DNA was quantified on a NanoDrop 1000 Spectrophotometer (Thermo Fisher Scientific, Waltham, MA, USA). The presence of NRTI drug-resistance mutations in the vectors were confirmed with Sanger sequencing using the BigDye Terminator v3.1 Cycle Sequencing Kit (Thermo Fisher Scientific, Waltham, MA, USA; Cat. No. 4337455) and ABI PRISM 3100 Genetic Analyzer (Thermo Fisher Scientific, Waltham, MA, USA). Sequencing chromatograms were analyzed using Geneious (Dotmatics, Boston, MA, USA) and the consensus sequences submitted to the Stanford HIV Drug Resistance Database [21] to confirm the presence of the target mutation(s).

For unsuccessful SDM reactions, an alternative approach, described by Yang et al., was followed using two sets of overlapping primers [22]. For each mutation, two reactions were performed. For the first reactions, the forward mutagenesis primer and AmpR1 reverse primer were used, while the reverse mutagenesis primer and AmpF1 forward primer were used (Appendix A). The reactions were constructed as follows: 12.5 μL Q5 Hot-Start High-Fidelity 2x Master Mix (New England Biosciences, USA; Cat. No. M0494S), 1.25 μL forward primer (10 μM), 1.25 μL reverse primer (10 μM), 1.0 μL plasmid DNA (25 ng/μL), and 9.0 μL nuclease-free water. Thermocycling commenced as follows: initial denaturation at 98 °C for 30 s; 25 cycles of denaturation at 98 °C for 10 s, annealing at 50–72 °C (primer specific) for 30 s, elongation at 72 °C for 10 min (50 s/kb); final extension at 72 °C for 2 min; and hold at 4 °C. The plasmid template DNA was digested by the addition of 0.2 μL *Dpn*I (20 U/μL; Thermo Fisher Scientific, Waltham, USA; Cat. No. ER1702) to each reaction and incubation at 37 for 2 °C hours. DH5α bacterial cells were transformed with 2.5 μL of each of the two reactions and the subsequent steps were followed as described previously. Selected mutations were also introduced into PSVs that contained a section of the *reverse transcriptase* from HIV-1 subtype B and C laboratory-adapted strains following the above SDM procedures.

To produce PSVs, HEK293T cells were plated in 10 mL of complete DMEM at 8 × 10^6^ cells/m Lin 10 cm Nunclon™ Delta transfection dishes (Thermo Fisher Scientific, Waltham, MA, USA; Cat. No. 150318) and incubated overnight at 37 °C under 5% CO_2_ in a humidified incubator. The transfection mixtures were prepared to contain the following: 1.00 μg HIV-1 gag–pol expression vector (mutant or wild-type), 0.25 μg pHEF-VSVG, 1.5 μg pCSFLW, 50 μL DMEM without additives, and 8.25 μL PEI “max” (1 mg/mL pH 7.0; PolySciences, Warrington, FL, USA; Cat. No. 24765-1). The transfection mixtures were incubated at room temperature for 20 min before adding it drop-wise to the plated cells. The plates were incubated for 48 h at 37 °C under 5% CO_2_ in a humidified incubator. The supernatants containing the PSVs were collected, filtered through a 0.45-micron syringe filter, aliquoted, and stored at −80 °C.

PSV titration was performed by preparing eight, two-fold serial dilutions of supernatant in 50 μL of complete DMEM in Nunc™ Edge™ 96-well culture plates (Thermo Fisher Scientific, Waltham, USA; Cat. No. 167542). After the addition of HEK293T cells (2 × 10^4^ cells/50 μL), the plates were incubated for 48 h at 37 °C under 5% CO_2_ in a humidified incubator. The expression of firefly luciferase, indicative of PSV infection, was assessed using the Bright-Glo™ Luciferase Substrate (Promega, Madison, WI, USA; Cat. No. E263B). For this, 100 μL of substrate was added to the wells of the 96-well culture plate and incubated for two minutes at room temperature in the dark. The well contents were mixed and transferred to the corresponding wells of a white 96-well plate. Bioluminescence was quantified on the GloMax^®^ Explorer Multimode Microplate Reader (Promega, Madison, WI, USA) in relative light units (RLU). PSV dilutions that produced a bioluminescence of 1 × 10^6^ RLU were used in subsequent in vitro phenotypic assays as a standardized PSV input.

### 2.7. Preparation of Mutant Pseudoviruses in Laboratory-Adapted Strains

Viral RNA was extracted from viral culture supernatants using the QIAamp Viral RNA Mini Kit (QIAGEN, Hilden, Germany; Cat. No. 52906) according to the manufacturer’s instructions. The viral RNA was reverse transcribed using the SuperScrip™ III First-Strand Synthesis System for RT-PCR (Thermo Fisher Scientific, Waltham, MA, USA; Cat. No. 18080-51) in conjunction with the Outer Reverse Primer (10 μM; 5′-GCTTGGATGCACACTAAATTTTCC-3′) according to the manufacturer’s instructions.

The first-round PCR reactions were performed using the Expand™ High-Fidelity PCR System (Merck, Darmstadt, Germany; Cat. No. EHIFI-RO) according to the manufacturer’s instructions and contained the following: 5 μL PCR Buffer (10×), 5 μL Outer Forward Primer (10 μM, 5′-CCATGGCTATTTTTTGCACTGC-3′), 5 μL Outer Reverse Primer (10 μM), 1 μL PCR-Grade Nucleotide Mix (10 mM of each dNTP), 0.75 μL Enzyme Mix (3.5 U/μL), 30.75 μL water, and 2.5 μL cDNA reaction. Thermocycling commenced as follows: initial denaturation at 94 °C for 2 min; 10 cycles of denaturation at 94 °C for 15 s, annealing at 55.5 °C for 30 s, and elongation at 72 °C for 1 min 10 s (45 s/kb); 20 cycles of denaturation at 94 °C for 15 s, annealing at 55.5 °C for 30 s, and elongation at 72 °C for 1 min 10 s (+5 s per successive cycle); final extension at 72 °C for 7 min; and hold at 4 °C.

The second-round PCR reactions were performed using the Platinum™ SuperFi II Green PCR Master Mix (Thermo Fisher Scientific, Waltham, MA, USA; Cat. No. 12369010) with primers that introduced *Pvu*I and *Hpa*I restriction sites at the 5′- and 3′-end of the amplicon, respectively. The reactions contained the following: 25 μL Platinum™ SuperFi II Green PCR Master Mix (2×), 2.5 μL Inner Forward Primer (10 μM; 5′-CGATCGAAACTGTACCAGTAAAATTAAAGC-3′), 2.5 μL Inner Reverse Primer (10 μM; 5′-GTTAACTGTTTTACATCATTAGTGTGGG-3′), 10 μL SuperFi II GC Enhancer (5×), 9 μL water, and 1.0 μL first-round PCR reaction. Thermocycling commenced as follows: initial denaturation at 98 °C for 30 s; 35 cycles of denaturation at 98 °C for 10 s, annealing at 56.4 °C for 30 s, and elongation at 72 °C for 1 min 10 s (45 s/kb); final extension at 72 °C for 5 min; and hold at 4 °C. The PCR reactions were analyzed on a 1% agarose gel in Tris-Acetate-EDTA (TAE) buffer (Thermo Fisher Scientific, Waltham, USA; Cat. No. B49) containing 0.35 μg/ mLethidium bromide (Bio-Rad, Johannesburg, South Africa; Cat. No. 1610433) to confirm the expected size of the amplicons (i.e., 1096 bp). The second-round PCR amplicon spanned from amino acid 4 (HXB2 nucleotide number 2560) to amino acid 368 (HXB2 nucleotide number 3656) in HIV-1 Reverse Transcriptase. The amplicons were gel-purified with the Zymoclean™ Gel DNA Recovery Kit (Zymo Research, Orange, CA, USA; Cat. No. D4001), and the eluted DNA was quantified on a NanoDrop 1000 Spectrophotometer (Thermo Fisher Scientific, Waltham, MA, USA) and sequenced. The absence of ARV drug-resistance mutations in the wild-type strains was confirmed with Sanger sequencing and submission to the Stanford HIV Drug Resistance Database [21].

The purified amplicons were cloned into the PCR-XL-2-TOPO vector using the TOPO™ XL-2 Complete PCR Cloning Kit (Thermo Fisher Scientific, Waltham, USA; Cat. No. K8050-10) according to the manufacturer’s instructions. Bacterial transformation, culture, and plasmid DNA extraction were performed as described previously. The plasmid DNA from the resulting clones were sequenced as described previously to confirm the identity of their inserts. After sequence confirmation, the TOPO clones were subjected to double digests: 0.5 μL *Pvu*I (20 U/μL; New England Biosciences, USA; Cat. No. R3150S), 0.5 μL *Hpa*I (5 U/μL; New England Biosciences, USA; Cat. No. R0105S), 1 μL T4 polynucleotide kinase (10 U/μL; New England Biosciences, USA; Cat. No. M0201S), 7 μL rCutSmart™ Buffer (10×), 1 μg plasmid DNA, and nuclease-free water to 70 μL final volume. Reactions were incubated at 37 °C for 1 h. The HIV-1 gag–pol expression vectors were also subjected to double-digests, similar to the TOPO clones, but the reactions contained 1.0 μL recombinant Shrimp Alkaline Phosphatase (1 U/μL; New England Biosciences, USA; Cat. No. M0371S) instead of T4 polynucleotide kinase. Reactions were incubated at 37 °C for 1.5 h, followed by 65 °C for 5 min to deactivate the phosphatase.

The inserts from the TOPO clones were then sub-cloned into the HIV-1 gag–pol expression vectors (i.e., p8.9NSX+ for subtype B strains, p8.9MJ4 for subtype C strains) using the Quick Ligation™ Kit (New England Biosciences, USA; Cat. No. M2200S). The reactions contained the following: 10 μL Quick Ligase Reaction Buffer (2×), 17 ng insert, 183.5 ng vector backbone, 1 μL Quick Ligase Enzyme, and nuclease-free water to 20 μL final volume. Reactions were incubated at room temperature for 5 min and 5 μL of each reaction was used for the transformation of DH5α bacterial cells, as described previously. The plasmid DNA from the resulting clones were sequenced to confirm the identity of their inserts. Consensus sequences were submitted to the Stanford HIV Drug Resistance Database [21] to confirm the absence of ARV drug-resistance mutations. NRTI drug-resistance mutations of interest were introduced into the clones by SDM (Appendix A), and PSVs were produced and tittered, as previously described.

### 2.8. In Vitro Phenotypic ISL Susceptibility Testing

An in vitro single-cycle phenotypic assay was used to assess the susceptibility of the PSVs to ISL [23,24,25], with some modification. Briefly, eleven duplicate, 3-fold serial dilutions of ISL in 50 μL complete DMEM were prepared in the wells of Nunc™ Edge™ 96-well culture plates (Thermo Fisher Scientific, Waltham, USA; Cat. No. 167542) with concentrations ranging from 500 nM to 0.008 nM. Two wells received complete DMEM only and functioned as the no-drug control. In addition, 3TC was included as a positive inhibition control over concentrations ranging from 100 μM to 0.002 μM. HEK293T cells were prepared in complete DMEM at 4 × 10^5^ cells/mL, PSVs were added to the cells at half the standardized dilution, and 50 μL of the cell/virus mixture was added to the wells of the plates. The plates were incubated for 48 h at 37 °C under 5% CO_2_ in a humidified incubator, after which the expression of firefly luciferase in the wells of the plates was assessed and quantified as previously described. All PSVs were assessed in three or more independent assays.

The percentage of viral activities were calculated for each PSV over the ISL concentration range that was tested, relative to the no-drug control. The inhibitory concentration-50s (IC_50_) were calculated for each PSV using the FORECAST function in Microsoft Excel (Redmond, OK, USA). This formula assumes a linear correlation between ISL concentration and percentage of viral activity. The degree of ISL susceptibility was expressed as fold-change (FC) in IC_50_, relative to the wild-type PSV IC_50_. The assay’s FC technical cut-off (TCO) for ISL was determined at the 99th percentile of the wild-type PSV IC_50_ values, determined over multiple repeat assays. The degrees of ISL resistance were set at increasing increments of the TCO value.

### 2.9. Statistical Methods

Data were evaluated for normality using the Shapiro–Wilk test, and significance using the Kruskal–Wallis test. A post hoc analysis was performed using Dunn’s multiple comparisons test to compare the median equality of the wild-type PSVs and mutant PSVs. The same statistical methods were used for intra- and inter-subtype comparisons.

## 3. Results

### 3.1. Sequence Database

#### 3.1.1. Regimens

Most sequences were obtained from patients failing a ritonavir-boosted protease inhibitor-based (PI/r) second-line regimen (70.89%, *n* = 5493/7749), while 12.67% (*n* = 982/7748) and 0.84% (*n* = 65/77498) were obtained from patients failing non-nucleoside reverse transcriptase inhibitor (NNRTI) or integrase strand-transfer inhibitor (INSTI)-based regimens, respectively. For 0.74% (*n* = 57/7748) of sequences, the information on the ARVs in the regimens were incomplete, while for 14.87% (*n* = 1152/7748) of sequences, no treatment information was provided. Details of the current and prior treatment are shown in Appendix A.

#### 3.1.2. NRTI-Resistant Mutations

Among the 7749 sequences, NRTI drug-resistance mutations were observed in 71.2% (*n* = 5521) of the sequences. The 20 most prevalent combinations (Appendix A) and individual single NRTI drug-resistance mutations that constitute the combinations (Appendix A) were identified. The M184V mutation was the most prevalent mutation, both as a single mutation and in combination with other NRTI drug-resistance mutations. The M184V mutation was observed in each of the 20 most prevalent mutation combinations. The thymidine analogue mutations (TAMs: M41L, D67N, K70R, T215FY, and K219QE) were more prevalent in combination with other mutations than in isolation. This was also observed for the non-TAM mutations (i.e., K70ET, L74IV, and Y115F) and for K65R. Sequences containing TAM-2 pathway mutation combinations (i.e., D67N/K70R/T215F/K219QE) were more prevalent (4.0%, *n* = 244/5521) than sequences containing TAM-1 pathway mutation combinations (i.e., M41L/L210W/T215Y) (1.9%, *n* = 109/5521). However, all TAM-1 pathway mutation combinations lacked the L210W mutation, and only the M41L+T215F or M41L+T215Y combinations were observed. We also observed the TAM-2 pathway mutation combination where only D67N+K07R was present. The remaining TAM-2 pathway mutation combinations contained at least three TAM-2 mutations. The A114S mutation, shown to reduce ISL susceptibility, either in isolation or in combination with the M184V mutation [16], was not observed in any of the sequences.

### 3.2. In Vitro Phenotypic ISL Susceptibility Testing

#### 3.2.1. TCO for ISL and Assay-Defined Classifications

To establish a meaningful threshold for the interpretation of the FC values of the HIV-1 subtype C mutant PSVs, a TCO was derived from the ISL IC_50_ values obtained from multiple independent in vitro phenotypic assays (*n* = 41) of the wild-type HIV-1 subtype C p8.9MJ4 PSV against ISL. The TCO was established at the 99th percentile (18.26 nM) of the mean IC_50_ (8.32 ± 4.99 nM S.D.), which translated to a TCO of 2.2 FC (Figure 1A,B). Thus, PSVs with a FC > 2.2 were considered to have a reduced susceptibility to ISL. Classifications of the degrees of reduced susceptibility or resistance were set at increments of the TCO as follows: FC < 2.2 = Susceptible, 2.2 ≤ FC ˂ 4.4 = Potential Low-Level Resistance, 4.4 ≤ FC ˂ 6.6 = Low-Level Resistance, 6.6 ≤ FC ˂ 8.8 = Intermediate-Level Resistance, FC ≥ 8.8 = High-Level Resistance. Similarly, for the interpretation of the FC values of the HIV-1 subtype B mutant PSVs, a TCO was established at the 99th percentile (15.29 nM) of the mean IC_50_ (7.91 ± 5.51 nM S.D.) of the wild-type HIV-1 subtype B p8.9NSX+, from multiple independent in vitro phenotypic assays (*n* = 13). The TCO was determined to be at an FC cut-off of 1.9 FC (Figure 1C,D). Classifications of the degrees of reduced susceptibility or resistance were set at increments of the TCO as follows: FC < 1.9 = Susceptible, 1.9 ≤ FC ˂ 3.8 = Potential Low-Level Resistance, 3.8 ≤ FC ˂ 5.7 = Low-Level Resistance, 5.7 ≤ FC ˂ 7.6 = Intermediate-Level Resistance, and FC ≥ 7.6 = High-Level Resistance.

#### 3.2.2. ISL Susceptibility of Single NRTI Drug-Resistance Mutations

PSVs (*n* = 16) containing single NRTI-resistant mutations were tested in vitro for phenotypic susceptibility to ISL. The majority of the PSVs (75%, *n* = 12) with single NRTI-resistant mutations (i.e., M41L, K65R, D67N, K70E/R/T, L74I, A114S, Y115F, T215Y, and K219E/Q) were fully susceptible to ISL (FC < 2.2; Figure 2). Three PSVs with single NRTI-resistant mutations showed potential low-level resistance to ISL: A62V = 2.79 (IQR 1.81–4.91) FC, L74V = 2.61 (IQR 0.83–4.13) FC, and T215Y = 3.12 (IQR 2.49–3.65). The PSV containing the M184V mutation was on the cusp of potential- and low-level resistance (FC = 4.63, IQR 3.27–5.41). It is important to note that the classifications of ISL susceptibility for several of the PSVs with single NRTI-resistant mutants (i.e., M41L, A62V, L74V, A114S, Y115F, M184V, and K219E) spanned over two or three classifications due to the variability in the assay. However, none of the mutations caused intermediate or high-level resistance to ISL.

A Kruskal–Wallis test was performed to assess the statistical significance in FC differences to the wild-type MJ4 PSV, which revealed a significant difference (*p* < 0.0001). A Post hoc Dunn’s multiple comparisons test showed that statistically significant differences were observed only for PSVs with the single M184V (*p* = 0.006) and T215Y (*p* = 0.025) mutants. However, although their changes in FC were significantly higher than the wild-type PSV, these mutations did not dramatically affect THE ISL susceptibility.

Published phenotypic data on ISL resistance were only available for a few of the NRTI-resistant mutations (i.e., M41L, K65R, L74I/V, A114S, M184V, and T215Y) (Appendix A). However, for the PSV with the L74V single mutant, we observed a median FC value of 2.61 (IQR 0.83–4.13), which was 13-fold higher than the reported value in the literature [8,10] (i.e., FC = 0.2). Since the PSV with this mutation also showed a high level of variation in IC_50_ in our in vitro assay (Figure 2, Appendix A), we generated and tested strain-derived PSVs with this mutation to ascertain whether the observations were strain and/or subtype (i.e., B or C)-related. All the strain-derived mutant PSVs that contained L74V were susceptible to ISL, with ISL FC values well below their corresponding wild-type references, irrespective of the subtype (Figure 3; Appendix A).

Inter-subtype and intra-subtype analyses were performed using the Kruskal–Wallis test. The inter-subtype comparison specifically scrutinized the IC_50_ values (i.e., not FC values) since no PSV could be used as a representative (or reference) for both subtypes (Figure 4A,B). There were no significant differences between the respective L74V mutants and their strains (subtype B: *p* = 0.348; subtype C: *p* = 0.074). The intra-subtype analysis, which compared the median FC values of L74V strains, also revealed no significant differences (*p* > 0.05) between any of the L74V-WT/strain mutants (Figure 4B, Appendix A).

#### 3.2.3. ISL Susceptibility of Combined NRTI Drug-Resistance Mutations

Overall, PSVs with combinations of two or more NRTI-resistant mutations were less susceptible to ISL (Figure 5) than those with single NRTI-resistant mutations (Figure 2). Regarding PSVs containing non-TAMs only, one combination (i.e., A62V/K65R/M184V) showed full susceptibility to ISL (FC = 1.42, IQR 1.17–1.62; Figure 5, Appendix A). Other PSVs containing combinations with K65R or K70E showed potential low-level resistance to ISL. The PSV with A62V/M184V was on the cusp of low- to intermediate-level ISL resistance (FC = 6.28, IQR 5.36–7.79). Although PSVs with combinations containing L74V (i.e., L74V/M184V, L74V/Y115F/M184V) showed low-level ISL resistance (i.e., FC = 2.73, IQR 1.28–3.90; FC = 4.28, IQR 3.82–5.30; Figure 5, Appendix A), the PSV with L74I/M184V showed high-level resistance (FC = 9.46, IQR 5.13–12.45; Figure 5, Appendix A). This was unexpected, as the PSV with the single L74I or M184V mutations were susceptible or showed a low level of resistance (L74I: FC = 0.84, IQR 0.73–0.94; M184V: FC = 4.63, IQR 3.27–5.41, Figure 2, Appendix A). The PSV with the A114S/M184V combination showed an extremely high level of resistance (FC > 60, IQR 0) to ISL. This was another unexpected result since the PSV with the single A114S mutation was susceptible to ISL (FC = 2.17, IQR 1.69–2.54, Figure 2, Appendix A), and the PSV with the single M184V mutation had a low level of resistance (Figure 2).

The PSVs that contained combinations with TAM-1 mutations showed potential low- to high-level ISL resistance (Figure 5). The combinations of M41L/M184V (FC = 4.20, IQR 3.54–5.17) and M41L/M184V/T215Y (FC = 4.36, IQR 2.51–5.32) showed potential low-level resistance to ISL, while the combination of M41L/M184V/T215F (FC = 9.26, IQR 7.42–13.98) showed a high level of resistance. The M184V/T215Y combination showed an intermediate level of ISL resistance (FC = 5.98, IQR 3.81–10.17). The PSVs that contained combinations with TAM-2 mutations generally showed higher levels of ISL resistance. Three combinations (i.e., K65R/M184V/K219E, D67N/K70E/M184V, and D67N/K70R/M184V) showed low-level ISL resistance (median FC 4.82–5.78).

Interestingly, the addition of K70E or K70R seemed to sensitize the D67N/M184V combination to ISL as they were more susceptible (FC 5.40–5.78) than the D67N/M184V combination (FC = 9.60, IQR 5.41–12.95), which showed an intermediate level of resistance. When three or more TAM-2 mutations were present, a high level of resistance to ISL was observed: D67N/K70R/M184V/K219E (FC = 15.52, IQR 12.39–17.77), D67N/K70R/M184V/K219Q (FC = 18.00, IQR 16.75–23.38), and D67N/K70R/M184V/T215F/K219E (FC = 18.94, IQR 12.45–25.36).

A Kruskal–Wallis test performed to assess the statistical significance in FC differences to the wild-type MJ4 PSV (Appendix A) revealed a significant difference (*p* < 0.0001). A post hoc Dunn’s multiple comparisons test showed that 9 out of 21 PSVs showed a significant difference in the median FC (*p*-value < 0.05), 5 of which were those containing TAM-2 mutations. None of the combinations that contained K65R showed a significantly higher median fold-change compared to the wild-type MJ4.

## 4. Discussion

Considering the integral role of NRTIs in all cART regimens, the development and exploration of novel NRTIs that are capable of effectively addressing prevailing NRTI-resistant viral variants is needed. ISL, a novel NRTtI with a unique mechanism of action, has an extended half-life in vivo and a characteristic resistance profile. Data on ISL phenotypic resistance is limited, particularly for HIV-1 non-B subtypes. To gain insight into the potential value of ISL as a treatment option in NRTI-experienced PLWH with HIV-1 subtype C, this study assessed the in vitro phenotypic drug susceptibility of HIV-1-like subtype C PSVs with common NRTI-resistant mutations to ISL. Prevalent NRTI-resistant mutations were identified from a routine genotypic drug resistance testing database, and PSVs containing the NRTI-resistant mutations were generated and tested for susceptibility to ISL in vitro. Phenotypic data for most PSVs with single or combinations of mutations were in agreement with the limited published phenotypic data available. No single NRTI-resistant mutation caused intermediate or high-level resistance to ISL. The highly prevalent M184V mutation conferred only a low level of resistance to ISL, whereas the K65R mutation was found to cause potential hyper-susceptibility to ISL. PSVs with combinations of NRTI-resistant mutations, all containing the M184V mutation, were generally less susceptible to ISL than those with single NRTI-resistant mutations. Combinations with the K65R, L74V, and K70E non-TAMs were either susceptible or had potential low-level resistance to ISL. Combinations with TAM-1, and particularly TAM-2 mutations, showed low- to high-level resistance to ISL. Both the type and number of TAMs impacted on the degree of ISL resistance.

The genotypic drug resistance testing database that was employed for this study included sequences from routine drug resistance testing performed during 2016–2020. The South African National Treatment Guidelines recommended cART initiation on TDF with either FTC or 3TC and efavirenz (EFV) [26]. PLWH, failing this first-line regimen, were switched to a protease inhibitor (PI)-based second-line regimen containing either AZT or abacavir (ABC) with 3TC and either ritonavir(r)-boosted lopinavir (LPV/r) or atazanavir (ATV/r). In May 2019, the guidelines changed to replace EFV in the first-line regimen with dolutegravir (DTG), an INSTI [27]. The majority of sequences were obtained from PLWH, managed under the 2015 guidelines, with a minority managed under the 2019 guidelines. Routine genotypic drug resistance testing for PLWH failing cART was only recommended for those failing a PI-based second-line regimen. As such, the majority of sequences that were scrutinized in this study were from PLWH who failed a PI-based regimen that contained AZT or ABC, and 3TC as NRTIs in their current regimens, and with d4T, FTC, or TDF in their previous regimens. This was reflected in their drug resistance profiles as we observed TAM-1 (i.e., M41L and T215Y) and TAM-2 (i.e., D67N, K70R, T215F, and K219EQ) mutations, typically associated with the sub-optimal use of AZT or d4T [28], as well as the L74IV and Y115F mutations which are typically associated with the sub-optimal use of ABC [29]. The M184V mutation was most prevalent and is typically selected for the sub-optimal use of either 3TC or FTC [30,31]. The presence of this mutation among the sequences is likely a result of the sub-optimal use of 3TC/FTC (XTC) during first-line cART and the sub-optimal use of 3TC during second-line cART. The presence of the K65R mutation is most likely a consequence of the sub-optimal use of TDF [32,33,34] during first-line cART, but could also be a result of the sub-optimal use of ABC during second-line cART [29]. Mutations at K70 (i.e., E/R/T) were also observed, likely as a consequence of the sub-optimal use of ABC or TDF [29,34]. The A62V mutation, usually observed in combination with other NRTI-resistant mutations [35], was also observed in two combinations (i.e., A62V/M184V and K65R/A62V/M184V) among the sequences.

A South African National Survey showed the M184V- and K65R NRTI-resistant mutations to be highly prevalent (78.2% and 57.5%, respectively) among PLWH who were failing a TDF-based regimen [36]. This highly prevalent M184V NRTI-resistant mutation typically decreases XTC susceptibility by 100- to 1000-fold in in vitro settings [30,31]. On the contrary, the M184V mutation is approximately three-fold more susceptible to AZT [37] and causes hyper-susceptibility to TDF [38]. Our data showed that this mutation conferred potential low-level resistance to ISL (FC = 4.63, IQR 3.27–5.41). This is in agreement with previous reports from Kawamoto et al. (FC = 7.0) [10], Grobler et al. (FC = 5.0) [8], and Diamond et al. (FC = 4.7–12.0) [16]. However, this contrasts with the FC values reported by Ohrui et al. (FC = 45.5) [39] and Oliveira et al. (FC = 78.9) [40]. An in vivo animal study performed on rhesus macaques showed no viral breakthroughs when infected with an M184V-containing simian immunodeficiency virus (SIV) in the presence of ISL [41]. This implies that ISL would still be efficacious in the treatment of PLWH with M184V-containing viral variants, which is much needed due to the mutation’s high prevalence. When appearing alone, the K65R mutation has been reported to cause resistance to XTC, ABC, and TDF (8.4- and 8.8-fold) [42], but increases susceptibility to AZT [37,43]. Our data showed that K65R alone was significantly more susceptible to ISL (i.e., FC = 0.35, IQR 0.09–0.61) than the wild-type reference, possibly suggesting its hyper-susceptible nature. This is in agreement with the existing literature [8,10,16,44]. The mechanism of the hyper-susceptibility of K65R to ISL has been shown to be mainly through the decreased excision of ISL from the DNA terminus [44]. Both the K65R and M184V mutations cause a deficit in viral replication [38]. An in vitro passage study showed that the single K65R mutant rapidly reverted to the wild-type in the presence of ISL, whereas no reversion was noted for M184V [45]. This points towards a sustained efficacy for ISL on K65R-containing viral variants.

Separately and in combination, the K65R and M184V mutations are two of the most common NRTI mutations to develop in those infected with HIV-1 subtype C, who are failing on a first-line NNRTI-backbone regimen containing TDF and 3TC [46,47]. This was also observed in our routine genotypic drug resistance testing database, which showed that the K65R/M184V combination was the second highest in terms of prevalence (i.e., 2.35%). While K65R and M184V are antagonistic mutations and cause a viral fitness deficit [48], the Stanford HIV Drug Resistance Algorithm [21] (https://hivdb.stanford.edu/, accessed 29 March 2024) predicts the combination to confer high-level resistance to ABC and XTC, and intermediate resistance to TDF. Our data showed that this combination conferred potential low-level resistance to ISL. The K65R mutation potentially sensitized the M184V mutant to ISL as it led to a 2.35-fold reduction in the median FC. This increase in the ISL susceptibility of K65R+M184V has also been observed in previous studies [8,49]. Interestingly, such an increase in susceptibility with K65R+M184V has also been observed with AZT [48]. In addition, we observed that combinations that contained other NRTI mutations (i.e., A62V, K70T, Y115F, or K219E), in addition to K65R+M184V, were susceptible to or conferred only a potential low-level resistance to ISL. The presence of other NRTI-resistant mutations in combination with K65R+M184V did not cause an additional decrease in ISL susceptibility. This finding is significant, especially in the context of subtype C, as we observed that 13.11% (*n* = 724) of the sequences from PLWH failing second-line therapy, that contained NRTI-resistant mutations, contained the K65R+M184V mutations in combination with other mutations.

Although AZT or ABC were occasionally included in first-line treatment, AZT was one of the main NRTIs used in PI-based second-line treatment. AZT has a low barrier to the development of drug resistance and often selects for TAMs [50,51,52]. TAMs occur in two distinguishable, yet overlapping patterns [28,53,54]: Type 1 TAMs include M41L, L210W, and T215Y; Type 2 TAMs include D67N, K70R, T215F, and K219E/Q. As 18.2% (*n* = 4275) of patients in the routine genotypic resistance database received AZT in their current regimens, several Type-1 (M41L and T215Y) and Type-2 (D67N, K70R, T215F, and K219E/Q) TAMs were observed. The TAMs were observed either alone, or in combination with other NRTI drug-resistance mutations. Individually, most of the PSVs with single TAMs were susceptible to ISL, although the T215Y mutant conferred potential low-level resistance to ISL. According to the Stanford HIV Drug Resistance Algorithm (https://hivdb.stanford.edu/, accessed 29 March 2024), T215Y is predicted to be susceptible to XTC, whilst conferring potential low-level resistance (~1.5-fold) to ABC and TDF. The remaining TAMs are predicted to be susceptible to XTC, ABC, and TDF, with the exception of T215F, which is predicted to confer potential low-level resistance to ABC and TDF. Limited data are available on the susceptibility of single TAM-containing variants on ISL susceptibility. A previous study reported an FC ≤ 2.5 for three single TAMs (i.e., M41L, L74I, and T215Y) [16], which is in agreement with our findings for these single mutations.

Although individual TAMs did not affect ISL susceptibility in our study, the combination of TAMs with other NRTI drug-resistance mutations led to a decrease in ISL susceptibility in most cases. This was most evident for combinations with TAM-2 mutations. The combination of TAM-1 mutations (i.e., M41L+M184V+T215Y) did not decrease ISL susceptibility. However, M41L+M184V, with the addition of the T215F TAM-2 mutation, caused a significant decrease in ISL susceptibility. Furthermore, the addition of other TAM-2 mutations (i.e., K70R/K219EQ, K70R/T215F/K219Q) to D67N/M184V significantly decreased ISL susceptibility. On the contrary, the addition of K70E (a non-TAM) or K70R (a TAM-2) to D67N/M184V increased ISL susceptibility, relative to that of D67N/M184V. Brenner et al. [55] showed that the combination of three TAM-1 mutations (M41L+L210W+T215Y) conferred low-level resistance to ISL (FC = 2.1). However, in combination with M184V, it conferred high-level resistance to ISL (FC = 40) [55]. In addition, Diamond et al. [16] showed that four TAM-2 mutations together (D67N+K70R+T215F+ K219Q), in the absence of M184V, conferred only low-level resistance to ISL (3.8 FC). Previous studies have reported that the combination of M184V with TAMs can resensitize TAMs to TDF [56,57]. However, as we have shown in this study, the opposite seems to be true for ISL, and the mechanism(s) by which the combination of TAMs with M184V increases the phenotypic resistance to ISL is unclear. ISL functions primarily as an immediate chain terminator and less frequently as a delayed chain terminator [6]. The M184V mutation causes NRTI resistance through discrimination between the deoxynucleoside triphosphates (dNTPs) and the chain-terminating NRTIs before incorporation into the elongating DNA strand during reverse transcription [58]. TAMs, on the other hand, cause resistance by removing the NRTIs from the DNA terminus after incorporation [59]. We, therefore, hypothesize that the combination of the two mutations could (i) prevent immediate chain termination by ISL through the action of the M184V mutation, and (ii) excise ISL during the delayed chain termination through the action of the TAMs. However, the inclusion of the K65R mutation in this mix could potentially swing the scales (as hypothesized above). The interaction between ISL and these antagonistic mutations are likely complex, and these hypotheses would need to be tested in future studies to confirm their validity.

The A62V mutation has been observed mostly in conjunction with the K65R mutation, partially correcting the viral fitness deficit caused by the latter [60]. However, in this study, the A62V+K65R+M184V mutation was found to be susceptible to ISL, most likely due to the hyper-susceptible nature of K65R+M184V. Contrasting this, the absence of the K65R mutant had a notable effect, as the A62V+M184V combination conferred a low to intermediate level of resistance to ISL. Although this may be a point of concern, the prevalence of the A62V+M184V combination was observed in only 1.01% of sequences with NRTI drug-resistance mutations in the genotypic drug resistance database.

We observed three variants at position 70 (i.e., K70E, K70R, and K70T) with FC values below the TCO. Two of these (i.e., K70E and K70T) showed a significant hyper-susceptibility to ISL. When in combination with D67N+M184V, K70E and K70R seemed to sensitize the D67N/M184V mutation to ISL. However, the addition of more TAM-2 mutations to D67N/K70R/M184V significantly reduced ISL susceptibility.

The L74V mutation showed a potentially low level of resistance to ISL (FC = 2.61, IQR 0.83–4.13) in our study, which was higher than that reported by Grobler et al. (FC = 0.21 ± 0.07) [8] and Kawamoto et al. (FC = 0.4) [10]. Further evaluation of the L74V mutation in both subtypes B and C PSVs in this study did show potential hyper-susceptibility in one of the four HIV-1 subtype B PSVs (i.e., DS9-L74V), although this was not significant. However, this was not observed in the remaining subtype B or C PSVs, and although unexplored, we suggest that hyper-susceptibility may be strain-specific for this NRTI-resistant mutation. An additional mutation at position 74 (i.e., L74I) was observed at a low frequency among our sequences and did not affect ISL susceptibility by itself. However, in combination with M184V, it significantly reduced ISL susceptibility.

The A114S+M184V NRTI-resistant mutation was initially identified in a selection study which showed that it incurs a high level of resistance to ISL (24-fold) [45]. Similarly, another in vitro dose-escalation study reported the A114S+M184V combination to have an FC of 37.9 to ISL, as well as confer very high levels of resistance to XTC (3TC: FC > 368, FTC: FC > 990) [16]. Interestingly, both studies reported the hyper-susceptible nature of the dual mutation against TDF (50-fold [45]/0.03 FC [16]) and AZT (0.4 FC) [16]. Although the A114S mutation alone did not decrease the susceptibility to ISL in our study (i.e., 2.17, IQR 1.69–2.54), a significant reduction in ISL susceptibility (i.e., FC > 60) was observed for the A114S+M184V combination. However, this mutation is uncommon, as the Stanford HIV Drug Resistance Database [61,62] (https://hivdb.stanford.edu/, accessed 29 March 2024) depicted its frequency as < 0.01%. In addition, the A114S+M184V mutant combination was shown to cause a deficit in viral fitness, leading to a significantly decreased infectivity when compared to the wild-type [45]. Therefore, although the A114S+M184V combination is of concern as it severely impacts ISL susceptibility, its low prevalence in PLWH and its detrimental impact on the virus fitness is reassuring.

This study has its limitations. There is a possibility that more than one genotypic drug resistance test could have been performed for some patients. However, according to the South African National Antiretroviral Treatment Guidelines, an HIV-1 genotypic drug resistance test is only performed after confirmed virological failure with two or more consecutive viral loads ≥1000 copies/ mL on a TDF/3TC/DTG (TLD) regimen or a boosted PI-based regimen for two years with confirmed adherence [20]. However, resistance testing first requires authorization by a member of the National Third-line committee, a helpline consultant, or a nominated provincial expert. Since genotypic drug resistance is tightly controlled, it is unlikely that the genotypic drug resistance database could contain more than one result per patient. The different classifications of phenotypic resistance were placed at increments of the TCO and are not linked to clinical outcomes. These classifications merely served as a means to contextualize the degrees of ISL resistance in the assay that was employed. Additionally, the mean FC value of at least three independent screens for the resistance classifications of each mutant PSV was utilized. For some PSVs, the phenotypic responses were either on the cusp of two neighboring classifications or the standard deviations overlapped two or three neighboring classifications. However, despite this, this study’s results were mostly in agreement with the limited literature on phenotypic ISL resistance currently available. We also observed large variations in the FC values in the repeat assays for some of the single and combinations of NRTI-resistant mutations. This may also be strain-specific, since we did not observe similar variations in the strain-derived PSVs compared to the L74V-MJ4 PSV. However, these variations (up to ±1.7 FC) were greater than the assay variation for the wild-type MJ4 PSV (i.e., ±0.6 FC), suggesting that there may be some other mechanisms in place that impact on the interaction between ISL and certain mutated amino acids in the active site of Reverse Transcriptase. We argue that this is a plausible hypothesis, considering the unique mechanisms of the action of ISL. Lastly, since all the NRTI mutation combinations observed among the sequences in our database contained M184V, TAMs in the absence of M184V were not investigated in this study. However, the results of our study reflect on a “real world” setting and its implication on ISL as a treatment option following failure on AZT/XTC-containing regimens.

ARV resistance and cross-resistance is the Achilles heel of cART. As we have shown in this study, regimen choices may have an impact on the efficacy of novel ARVs, even those with novel mechanisms of action. This study implies that failure on a TDF/XTC-containing cART would possibly have a minimal impact on ISL efficacy in PLWH. However, due to the frequent selection of TAMs in conjunction with the M184V NRTI-resistant mutation, PLWH failing AZT/XTC-containing cART would most likely not benefit from ISL. With the shift from NNRTI-based first-line regimens (e.g., TDF/3TC/EFV) to INSTI-based regimens (e.g., TLD) [63], ISL would remain efficacious since prevalent NRTI-resistant mutations, selected for by TDF (i.e., K65R) and 3TC (M184V), would have a low impact on ISL susceptibility. Despite the effectiveness of PI-based second-line regimens that contain AZT/FTC, the selection of TAMs in the presence of M184V could negate the efficacy of ISL. The Nucleosides and Darunavir/Dolutegravir in Africa (NADIA) Trial demonstrated that TDF/3TC was non-inferior to AZT/3TC in second-line cART [64]. The data from this trial support maintaining TDF and 3TC at the time of a switch to second-line treatment. Due to the absence of AZT in such regimens, ISL could potentially be used effectively as an alternative switching option, or in subsequent third-line cART. However, the full treatment histories of PLWH should be taken into consideration prior to switching to ISL-containing regimens.

## Figures and Tables

**Figure 1 viruses-16-01888-f001:**
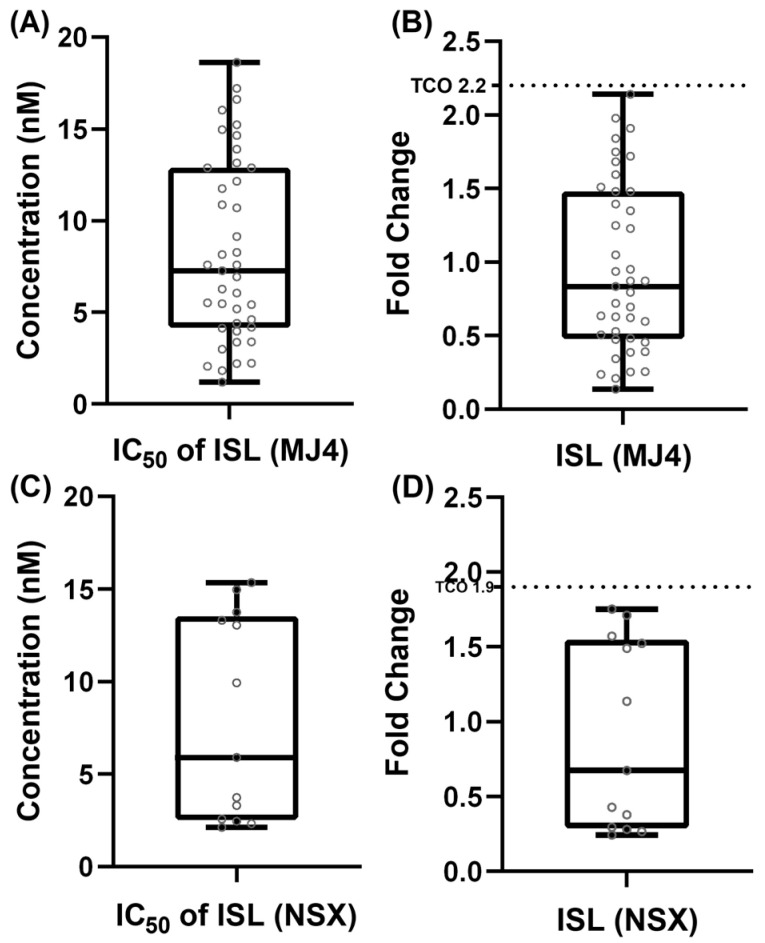
Variation in IC_50_ and fold-change of ISL against wild-type PSV. (**A**) Multiple independent in vitro assays (*n* = 41) were conducted to determine the IC_50_ of ISL against the wild-type HIV-1 subtype C virus (i.e., p8.9MJ4). The mean IC_50_ = 8.32 nM ± 4.99 nM (median IC_50_ = 7.27 nM, IQR 4.16–12.88). (**B**) Each IC_50_ value was divided by the mean IC_50_ value to determine the fold-change (FC). The mean FC was, therefore, 1.0 ± 0.6 FC (median FC = 0.87, IQR 0.54–1.55). Technical cut-off (TCO): obtained from the 99th percentile of the IC_50_ values. (**C**) Multiple independent in vitro assays (*n* = 13) were conducted to determine the IC_50_ value of ISL against the wild-type HIV-1 subtype B virus (i.e., p8.9NSX). The mean IC_50_ concentration was shown to be 7.91 nM ± 5.51 nM (median IC_50_ = 5.90 nM, IQR 2.58–13.32). (**D**) Each IC_50_ value was divided by the mean IC_50_ value to obtain a mean FC value of 1 (median FC = 0.74, IQR 0.33–1.68).

**Figure 2 viruses-16-01888-f002:**
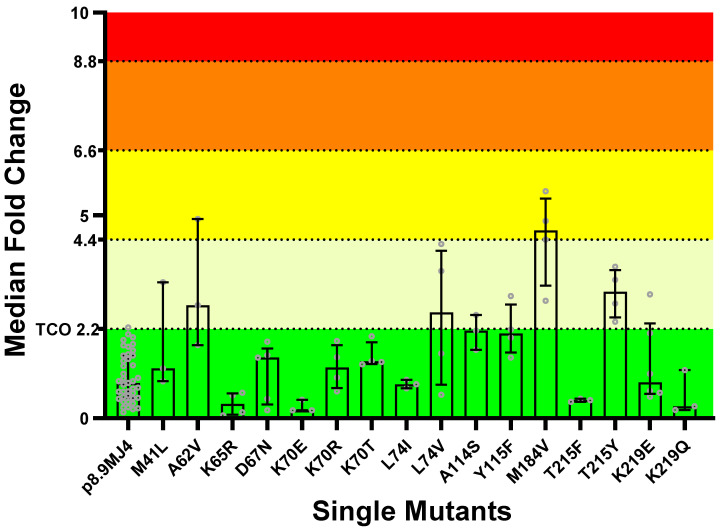
Fold-change in IC_50_ of single mutants in subtype C PSVs compared to the wild-type PSV. Following the in vitro phenotypic activity assays, the IC_50_ values for each mutant was compared against the mean IC_50_ of the MJ4 wild-type PSV, allowing the determination of fold changes. The 99th percentile of variation in the wild-type IC_50_ value, calculated to be 2.2 (TCO), served as the threshold for categorizing mutants as either susceptible or having a decreased susceptibility to ISL. Among the single mutants, M41L, K65R, D67N, K70E/R/T, L74I, A114S, Y115F, T215F, and K219E/Q demonstrated susceptibility to ISL. In contrast, A62V, L74V, and T215Y exhibited potential low-level resistance, and M184V exhibited potential-low- to low-level resistance. Susceptible (*n*), potential-low-level resistance (*n*), low-level resistance (*n*), intermediate resistance (*n*), and high-level resistance (*n*). Data are shown as median bar graphs with the IQR as error bars.

**Figure 3 viruses-16-01888-f003:**
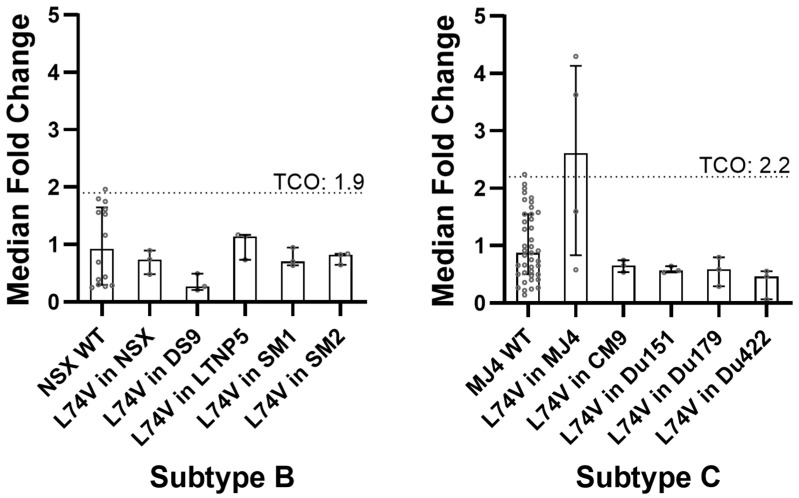
Single mutant L74V in wild-type HIV-1 subtype B and C laboratory-adapted strains. A phenotypic activity assay was conducted to determine whether ISL had similar potencies against the single mutant L74V in different wild-type strains and subtypes of HIV-1. IC_50_ values are expressed as median fold-change differences to the IC_50_ of the relevant control subtype, with the IQR as error bars. The TCO for each subtype is indicated on the graph by a dotted line.

**Figure 4 viruses-16-01888-f004:**
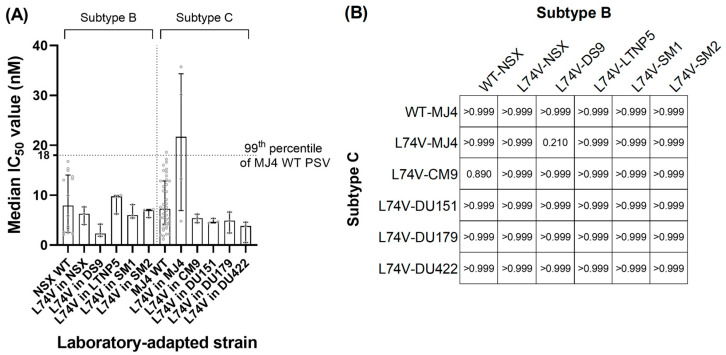
IC_50_ values of L74V in laboratory-adapted PSVs and inter-subtype Kruskal–Wallis test. Site-directed mutagenesis was performed to introduce the L74V mutant into the wild-type p8.9NSX and p8.9MJ4, and laboratory-adapted strains. (**A**) Median IC_50_ values of the L74V mutation in laboratory-adapted strains show that the LTNP5 PSV has the highest median IC_50_ of 9.76 nM (IQR 6.27–9.98). DS9 had the lowest median IC_50_ of 2.31 nM (IQR 1.78–4.20). Data are shown as median bar graphs with the IQR. (**B**) Inter-subtype non-parametric statistical analysis was performed using a Kruskal–Wallis multiple comparisons test. The grid shows the *p*-values of the comparisons in IC_50_ values. No significant differences (*p* > 0.05) were observed.

**Figure 5 viruses-16-01888-f005:**
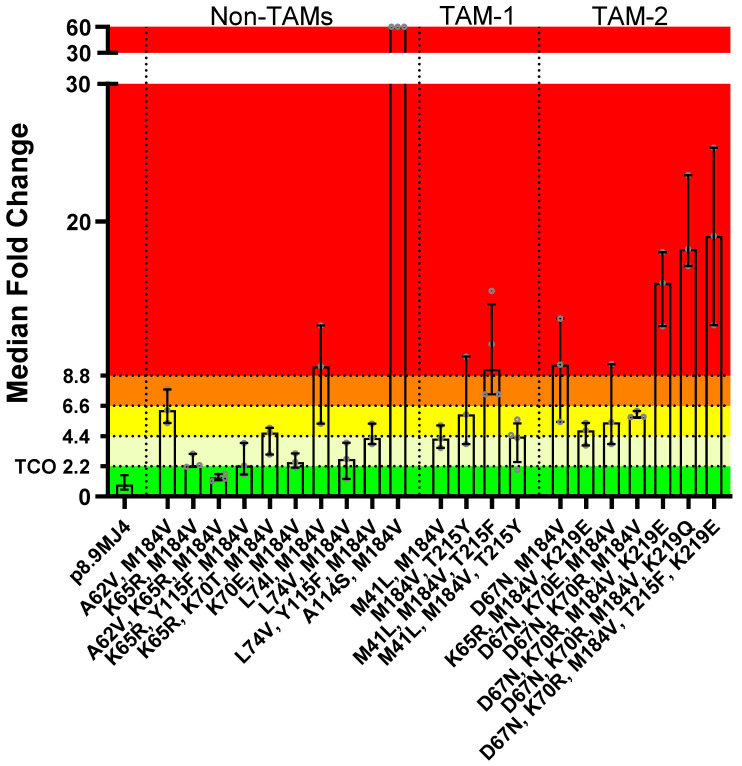
Fold-change in IC_50_ of mutation combinations in subtype C PSVs compared to the wild-type MJ4 PSV. Following the in vitro phenotypic activity assays, the IC_50_ value for each mutant was compared against the mean IC_50_ of the MJ4 wild-type PSV, allowing for the determination of fold changes. The 99th percentile of variation in the wild-type IC_50_ value, calculated to be 2.2 (TCO), served as the threshold for categorizing mutants as either susceptible or resistant to ISL. It was observed that the combination of NRTI mutations generally increased resistance to ISL. The A114S/M184V mutation combination showed a very high level of resistance to ISL. Its IC_50_ value was greater than the highest ISL concentration tested, and consequently, its FC value was > 60. Susceptible (*n*), potential-low-level resistance (*n*), low-level resistance (*n*), intermediate resistance (*n*), and high-level resistance (*n*). Data are shown as median bar graphs with the IQR as error bars.

## Data Availability

The original contributions presented in this study are included in the article/Appendix A; further inquiries can be directed to the corresponding authors.

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
