# Peer review of "Thymidine Analogue Mutations with M184V Significantly Decrease Phenotypic Susceptibility of HIV-1 Subtype C Reverse Transcriptase to Islatravir"

_viruses, 2024, doi:10.3390/v16121888_

Round 1

Reviewer 1 Report

Comments and Suggestions for Authors

In this manuscript “Thymidine Analogue Mutations with M184V Significantly Decrease Phenotypic Susceptibility of HIV-1 subtype C Reverse 3 Transcriptase to Islatravir”, the authors reported an in vitro inhibitory experiment of Islatravir (ISL), a novel HIV antiviral that could potentially serve in a long-acting regimen, against HIV-1 subtype C strains with different RT mutations. 

The manuscript is comprehensive and well-written. The findings add values to the current resistance study for this novel HIV antiviral. I have a few comments and suggestions. 

  1. Section 3.1.2. I assume all these sequences are from subtype C HIV-1. But do authors have any data from other subtypes?
  2. The resolutions of the figures are poor. Please use high resolution figures.
  3. Line 366. K70T has a FC of 1.5. It is truly an indication of hyper-susceptibility to ISL?
  4. Line 403. Please use nM instead of µM for the consistency. 
  5. I’m not sure how to interpret the differences the authors observed in Figure 4. They had the significant p value comparing WT-MJ4 vs L74V-DS9 probably because MJ4 WT had so many data points. I also did not see an explanation why L74V-MJ4 had such high variance in IC50 values. 
  6. A114S mutation has been reported as one of the major resistance mutation to ISL, but not in this study. Any explanation why the authors observed the differences?

Author Response

Reviewer 1

We thank the Reviewer for taking the time and making the effort to review our manuscript. Thank you for the valuable comments. Please see below for our responses to the comments.

Comment 1:

Section 3.1.2. I assume all these sequences are from subtype C HIV-1. But do authors have any data from other subtypes?

Response:

We focused specifically on HIV-1 subtype C as this is the prevalent circulating subtype in South Africa. Apart from the subtype-B lab-adapted strains that were also included in this study, we unfortunately do not have data on any other subtypes.

Comment 2:

The resolutions of the figures are poor. Please use high resolution figures.

Response:

We apologize for the poor resolution of the figures. The resolution seemed sufficient when the manuscript was initially submitted. However, we have now improved the resolution of the figures.

Comment 3:

Line 366. K70T has a FC of 1.5. It is truly an indication of hyper-susceptibility to ISL?

Response:

We have now reanalyzed the data with other statistical methods and do not consider K70T to be hypersusceptible.

Comment 4:

Line 403. Please use nM instead of µM for the consistency.

Response:

We have changed the IC50 values in the legend of Figure 4, as well as the scale of the y-axis of Figure 4a, from mM no nM.

Comment 5:

I’m not sure how to interpret the differences the authors observed in Figure 4. They had the significant p value comparing WT-MJ4 vs L74V-DS9 probably because MJ4 WT had so many data points.

Response:

We apologize for the lack of clarity. We performed an inappropriate statistical test which showed a statistical significance in the IC50 values of the L74V-DS9 PSV and the wild-type MJ4 PSV. We have now performed the appropriate statistical test, which showed no statistical difference between the two IC50 values.

Comment 6:

I also did not see an explanation why L74V-MJ4 had such high variance in IC50 values.

Response:

Regarding the high variation in IC50 values for the MJ4-L74V PSV, unfortunately we do not have an explanation for this. We have included the following hypothesis for this in the original Discussion.

See lines 678-687 in the original manuscript: “We also observed large variations in FC values in the repeat assays for some of the single and combinations of NRTI resistance mutations.”… “However, these variations (up to ±1.7 FC) were greater than the assay variation for the wild-type MJ4 PSV (i.e., ±0.6 FC), suggesting that there may be some other mechanisms in place that impact on the interaction between ISL and certain mutated amino acids in the active site of Reverse Transcriptase. We argue that this is a plausible hypothesis, considering the unique mechanisms of action of ISL.”.

Comment 7:

A114S mutation has been reported as one of the major resistance mutation to ISL, but not in this study. Any explanation why the authors observed the differences?

Response:

Regarding the A114S mutation reported as one of the major resistance mutations to ISL, Cilento et al. (2021) and Diamond et al. (2022) both reported that the A114S mutation alone only causes ~2-fold decrease in susceptibility to ISL (see Table S6). This is in line with our finding, where the A114S mutation alone caused a 2.17 (IQR 1.69 – 2.54) decrease in ISL susceptibility. Both authors have also observed a high level of ISL with A114S in the presence of M184V, as we have also (see Table S8).

Reviewer 2 Report

Comments and Suggestions for Authors

Byun et al. investigated the effects of mutations associated with nucleoside analog resistance on islatravir (ISL) susceptibility in HIV-1 group M subtype C. A considerable effort was made to identify the most common mutational patterns in subtype C reverse transcriptase that are associated with NRTI exposure. These changes were introduced either on-by-one or in various combinations into subtype C-based vectors, and ISL susceptibility was measured over a single round of infection in HEK293T cells. M184V, either alone or in combination with certain thymidine analog mutations, appeared to confer resistance to ISL, whereas K65R and possibly K70R and T215F appeared to individually confer ISL hypersusceptibility.

Major Comments

1)   The authors seem to be unaware that islatravir was formerly known as EFdA, which is an abbreviation of 4´-ethynyl-2-fluoro-2´-deoxyadenosine; similar chemical names have also been used in the past (see below). Thus, at various points in the manuscript, and especially in Tables S6 and S8, key publications relating to ISL resistance in subtype B HIV-1 are not cited, and the findings from these past works are not considered in the current study. A few examples:

Ohrui et al. 2'-Deoxy-4'-C-ethynyl-2-fluoroadenosine: a nucleoside reverse transcriptase inhibitor with highly potent activity against all HIV-1 strains, favorable toxic profiles and stability in plasma. Nucleic Acids Symp Ser (Oxf). 2006; (50):1-2. PMID: 17150787.

Ohrui, H. 2'-deoxy-4'-C-ethynyl-2-fluoroadenosine, a nucleoside reverse transcriptase inhibitor, is highly potent against all human immunodeficiency viruses type 1 and has low toxicity. Chem Rec. 2006; 6(3):133-43. PMID: 16795005.

Kawamoto et al. 2'-deoxy-4'-C-ethynyl-2-halo-adenosines active against drug-resistant human immunodeficiency virus type 1 variants.

Int J Biochem Cell Biol. 2008; 40(11):2410-20. PMID: 18487070.

Maeda et al. Delayed emergence of HIV-1 variants resistant to 4'-ethynyl-2-fluoro-2'-deoxyadenosine: comparative sequential passage study with lamivudine, tenofovir, emtricitabine and BMS-986001. Antivir Ther. 2014; 19(2):179-89. PMID: 24162098.

Oliveira et al. M184I/V substitutions and E138K/M184I/V double substitutions in HIV reverse transcriptase do not significantly affect the antiviral activity of EFdA. J Antimicrob Chemother. 2017 Nov 1;72(11):3008-3011. PMID: 28961903.

Also, a paper relevant to K65R and ISL hypersusceptibility, which contsins single-round cell culture data, is not cited:

Michailidis et al. Hypersusceptibility mechanism of Tenofovir-resistant HIV to EFdA. Retrovirology 2013 Jun 24:10:65. PMID: 23800377.

2)   There are serious problems with the way that the data are analyzed. The use of pairwise t-tests in Figures 2, 3, and 5 is inappropriate and susceptible to type I errors. All experiments that involve more than two strains or groups should be subjected to an ANOVA of the underlying IC50 values with an appropriate multiple comparisons test (for example, Tukey’s post-test). It would be  acceptable to LOG(10)-transform the IC50s in Figure 5, since the values cross multiple orders of magnitude. But, if the variation in the wild-type IC50 is too great to show a statistically-significant difference by ANOVA, then either there really is no measurable difference, or a more reliable assay needs to be used.

3)   A better description of the regression model is needed. Is the FORECAST function that was used a linear model? If so, how were the data chosen to fit the linear portion of the dose-response curve?

4)   Designations of potential low-level, low-level, intermediate resistance, etc... are completely arbitrary and can be misleading. It is acceptable to use general descriptors such as “highly resistant” or low-level-resistance” in the text, followed by an indication of fold change or some other measured parameter. But any attempt to set cutoffs for these categories, and in particular, their use in Figures 2 and 5 (as indicated by different colors) is invalid.

5)   More details are needed about the database and searches.  What search terms were used? What date was the database accessed on? Are these data publicly available? Does it have a name? Also, was Human Subjects approval given for this study? Which IRB(s) gave approval? Was informed consent obtained? Was there any effort to deal with the fact that some patients probably had multiple geneotypic tests during the study interval? This information should be included in the Methods and Materials.

Minor Comments

6)   The legend for Figure 4 should state that the indicated mutations were introduced into the MH4 background by site-directed mutagenesis.

7)   Typo on line 418: M814V.

Author Response

Reviewer 2

We thank the Reviewer for taking the time and making the effort to review our manuscript. Thank you for the valuable comments. Please see below for our responses to the comments.

Comment 1:

The authors seem to be unaware that islatravir was formerly known as EFdA, which is an abbreviation of 4´-ethynyl-2-fluoro-2´-deoxyadenosine; similar chemical names have also been used in the past (see below). Thus, at various points in the manuscript, and especially in Tables S6 and S8, key publications relating to ISL resistance in subtype B HIV-1 are not cited, and the findings from these past works are not considered in the current study. A few examples (were provided).

Response:

Regarding the more “historic”/initial work performed with ISL, we have now included most of the suggested references (i.e., those with phenotypic data) in Table S6 and the related text.

We have also included some information on the preliminary work on EFdA drug resistance in the Introduction.

Please see line 44: “Islatravir (ISL, EFdA or MK-85910) is a novel….”

Please also see lines 48 – 54: “Early studies on ISL showed an in vitro potency 10-fold higher than the approved NRTIs TAF, AZT, 3TC [8]. The common HIV-1 NRTI mutation M184V showed a nine-fold decrease in ISL susceptibility in HIV-1 subtype B NL4-3 [9]. However, the K65R, L74V, and Q151M mutations were more susceptible to ISL [8]. ISL was also shown to retain its potency against the multi-drug resistant HIV-1 strain A62V/V75I/F77L/ F116Y/Q151M [10], which is resistant to all current NRTIs except tenofovir [11].”

Comment 2:

Comment 2.1:

There are serious problems with the way that the data are analyzed. The use of pairwise t-tests in Figures 2, 3, and 5 is inappropriate and susceptible to type I errors. All experiments that involve more than two strains or groups should be subjected to an ANOVA of the underlying IC50 values with an appropriate multiple comparisons test (for example, Tukey’s post-test). It would be  acceptable to LOG(10)-transform the IC50s in Figure 5, since the values cross multiple orders of magnitude.

Response:

Regarding the use of inappropriate statistical methods, we have consulted a biostatistician at the University of the Witwatersrand and have now used the appropriate statistical tests to compare fold-change (FC) and IC50 values of the variants.

Please refer to lines 262 – 266: “Data were evaluated for normality using the Shapiro-Wilk test, and significance using the Kruskal-Wallis test. A post-hoc analysis was performed using Dunn’s multiple comparisons test to compare the median equality of the wild-type PSVs and mutant PSVs. The same statistical methods were used for intra- and inter-subtype comparisons.”

We have also updated the figures, supplementary tables and corresponding text in line with the results of these statistical methods.

Comment 2.2:

But, if the variation in the wild-type IC50 is too great to show a statistically significant difference by ANOVA, then either there really is no measurable difference, or a more reliable assay needs to be used.

Response:

Regarding the large variation in the ISL IC50 values of the wild-type variant, this is to be expected as the assay employed live cells, which is significantly more complex than, for example, an enzymatic assay for which one would most likely observe smaller variations. However ,the IC50 variation observed for ISL (i.e., 2.2-fold) is in line with variations in IC50s observed for other antiretroviral drugs in the same assay: Etravirine (3.6-fold1), Rilpivirine (2.6-fold1), Efavirenz (3.8-fold1), Nevirapine (2.8-fold1), Zidovudine (2.1-fold2), Stavudine (2.9-fold2) and Lamivudine (2.7-fold2). We therefore accept that the IC50 values of the wild-type and mutant variants reported in this manuscript is reliable for the assay that was employed.

References:

  1. Basson AE, Rhee SY, Parry CM, El-Khatib Z, Charalambous S, De Oliveira T, Pillay D, Hoffmann C, Katzenstein D, Shafer RW, Morris L. Impact of drug resistance-associated amino acid changes in HIV-1 subtype C on susceptibility to newer nonnucleoside reverse transcriptase inhibitors. Antimicrob Agents Chemother. 2015 Feb;59(2):960-71. doi: 10.1128/AAC.04215-14. Epub 2014 Nov 24. PMID: 25421485; PMCID: PMC4335849.
  2. Basson AE, Charalambous S, Hoffmann CJ, Morris L. HIV-1 re-suppression on a first-line regimen despite the presence of phenotypic drug resistance. PLoS One. 2020 Jun 18;15(6):e0234937. doi: 10.1371/journal.pone.0234937. PMID: 32555643; PMCID: PMC7302689.

Comment 3:

A better description of the regression model is needed. Is the FORECAST function that was used a linear model? If so, how were the data chosen to fit the linear portion of the dose-response curve?

Response:

Regarding the “regression model” that was used to calculate the IC50 values, we have now indicated that the FORECAST formula in Microsoft Excel assumes a linear relationship between ISL concentration and the in vitro response. This is not a regression model per se. However, since a linear relationship is observed between drug concentration and the response in a sigmoidal dose-response curve, this formula is appropriate to use to calculate the IC50 value. We have compared sigmoidal dose-response curve-fitted IC50 values with IC50 values obtained using the FORECAST formula, and the values were either the same or within 1-fold of each other. In addition, others have used the same method for calculating iC50 values (see references below). We therefore accept this method as an appropriate means to calculate the IC50 values presented in this manuscript.

References:

  1. Parry CM, Kohli A, Boinett CJ, Towers GJ, McCormick AL, Pillay D. Gag determinants of fitness and drug susceptibility in protease inhibitor-resistant human immunodeficiency virus type 1. J Virol. 2009 Sep;83(18):9094-101. doi: 10.1128/JVI.02356-08. Epub 2009 Jul 8. PMID: 19587031; PMCID: PMC2738216.
  2. Gupta RK, Kohli A, McCormick AL, Towers GJ, Pillay D, Parry CM. Full-length HIV-1 Gag determines protease inhibitor susceptibility within in vitro assays. AIDS. 2010 Jul 17;24(11):1651-5. doi: 10.1097/qad.0b013e3283398216. PMID: 20597164; PMCID: PMC2923069.

Please refer to lines 257 – 258: “This formula assumes a linear correlation between ISL concentration and percent viral activity.”

Comment 4:

Designations of potential low-level, low-level, intermediate resistance, etc... are completely arbitrary and can be misleading. It is acceptable to use general descriptors such as “highly resistant” or low-level-resistance” in the text, followed by an indication of fold change or some other measured parameter. But any attempt to set cutoffs for these categories, and in particular, their use in Figures 2 and 5 (as indicated by different colors) is invalid.

Response:

Regarding the misleading phenotypic cut-off values, we have clearly stated the following in the text of the manuscript of the original manuscript (see lines 671 – 674): “This study has its limitations. The different classifications of phenotypic resistance were placed at increments of the TCO and are not linked to clinical outcomes. These classifications merely served as a means to contextualize the degrees of ISL resistance in the assay that was employed.” Even without using these assay-specific technical cut-off values, the mutations impact on ISL susceptibility is evident in our data. ISL is still under clinical investigation and phenotypic clinical cut-off’s related to treatment outcomes of people living with islatravir resistant HIV variants have not been published. Without clear definitive guidelines on defining cut-off values in vitro phenotypic assays, we implemented a statistical approach (i.e., therefor arbitrary), based on variation in our assay, to define these in the context of the assay that was used in our study. Even if “It is acceptable to use general descriptors such as “highly resistant” or low-level-resistance” in the text, followed by an indication of fold change or some other measured parameter.”, there is not clear indication of where to “draw the line” between the two classifications. Using increments of the 99th percentile in IC50 value was the most logical approach to used, in our opinion. The data presented in our manuscript would be of great value to HIV clinicians and care givers, irrespective of where we “drew the lines” and especially considering the novelty of the ISL and the scarcity of published phenotypic drug resistance data on ISL.

Comment 5:

More details are needed about the database and searches.  What search terms were used? What date was the database accessed on? Are these data publicly available? Does it have a name? Also, was Human Subjects approval given for this study? Which IRB(s) gave approval? Was informed consent obtained? Was there any effort to deal with the fact that some patients probably had multiple geneotypic tests during the study interval? This information should be included in the Methods and Materials.

Response:

Regarding the genotypic drug resistance database that was employed, we have now included some additional information in the text.

Please refer to lines 102 – 107: “The Nation Health Laboratory Services (NHLS) is a South African governmental institution that performs various serological and pathological tests on patients in the South African national health sector. Tests are requested by health care providers and are performed with the patient’s consent. For this study, the use anonymized HIV-1 genotypic drug resistance data was approved by the University of the Witwatersrand Human Research Ethics Committee (protocol M221081, ethics number R14/49).”

Regarding multiple genotypic drug resistance test per patient, we have included the following under the study limitations: 

Please see lines: 640 – 649: “There is a possibility that more than one genotypic drug resistance test could have been performed for some patients. However, according to the South African National Antiretroviral Treatment Guidelines, an HIV-1 genotypic drug resistance test is only performed after confirmed virological failure with two or more consecutive viral loads ≥ 1000 copies/ml on a TDF/3TC/DTG (TLD) regimen or a boosted PI-based regimen for two years with confirmed adherence [20]. However, resistance testing first requires authorization by a member of the National Third-line committee, a helpline consultant, or a nominated provincial expert. Since genotypic drug resistance is tightly controlled, it is unlikely that the genotypic drug resistance database could contain more than one result per patient.”

Comment 6:

The legend for Figure 4 should state that the indicated mutations were introduced into the MH4 background by site-directed mutagenesis.

Response:

This has now been added.

Please see lines 398 – 399 (in Figure 4 legend): “Site-directed mutagenesis was performed to introduce the L74V mutant into the wild-type p8.9NSX and p8.9MJ4, and laboratory-adapted strains.”

Comment 7:

Typo on line 418: M814V.

Response:

This has now been corrected.

Please see line 415: “…as the PSV with the single L74I or M184V mutations were susceptible…”

Reviewer 3 Report

Comments and Suggestions for Authors

In this study, the authors investigate the activity of islatravir against subtype C sequences containing single and multiple NRTI resistance mutations. ACtivity of islatravir against subtype C HIV has not been previously reported in depth, and this study provides important insights.

There are some caveats to the study - for example he clinical cut-off for resistance has yet to be determined and as such the determinations of low, intermediate, high level resistance are somewhat theoretical, and the study uses recombinant viruses. Nevertheless, the data are convincing, and are rigorously reported. This study should be of significance to HIV researchers interested in ART and/or drug resistance.

Line 366 is somewhat confusing - K70T is reported to have 1.5-fold resistance but is clearly hypersusceptible. I am assuming that value should be 0.6?

The combination of K65R or M184V in combination with TAMs confers high-level resistance to islatravir. K65R and M184V are antagonistic to TAMS, and TAMs can be antagonistic to K65R (see studies from Gotte/Wainberg and Mellors). The mechanism by which 65r?184V and TAMs confer NRTI resistance are also very different, and it would be useful if the authors could speculate as to wht these combinations confer such high-levels of islatravir resistance.

K70T not hypersusceptible as stated on 366 Fold change not 1.5?

Author Response

Reviewer 3

We thank the Reviewer for taking the time and making the effort to review our manuscript. Thank you for the valuable comments. Please see below for our responses to the comments.

Comment 1:

Line 366 is somewhat confusing - K70T is reported to have 1.5-fold resistance but is clearly hypersusceptible. I am assuming that value should be 0.6?

Response:

We have now reanalyzed the data with other statistical methods and do not consider K70T to be hypersusceptible.

Comment 2:

The combination of K65R or M184V in combination with TAMs confers high-level resistance to islatravir. K65R and M184V are antagonistic to TAMS, and TAMs can be antagonistic to K65R (see studies from Gotte/Wainberg and Mellors). The mechanism by which 65r?184V and TAMs confer NRTI resistance are also very different, and it would be useful if the authors could speculate as to wht these combinations confer such high-levels of islatravir resistance.

Response:

Regarding the possible mechanism behind the increased ISL phenotypic resistance of PSVs with TAM + M184V, we are only able to speculate.

Lines 573 – 588: “Previous studies have reported that the combination of M184V with TAMs can resensitize TAMs to tenofovir (Wolf et al., 2003; Harrigan et al., 2002). However, as we have shown in this study, the opposite seems to be true for ISL and the mechanism(s) by which the combination of TAMs with M184V increases phenotypic resistance to ISL is unclear. ISL functions primarily as an immediate chain terminator and less frequently as a delayed chain terminator (Michailidis et al., 2009). The M184V mutation causes NRTI  resistance through discrimination between the deoxynucleoside triphosphates (dNTPs) and the chain-terminating NRTIs before incorporation into the elongating DNA strand during reverse transcription (Sluis-Cremer et al., 2000). TAMs, on the other hand, cause resistance by removing the NRTIs from the DNA terminus after incorporation (Gotte at al., 2007). We therefore hypothesize that the combination the two mutations could (i) prevent immediate chain termination by ISL through the action of the M184V mutation, and (ii) excise ISL during the delayed chain termination through the action of the TAMs. However, the inclusion of the K65R mutation in this mix could potentially swing the scales (as hypotheses above). The interaction between ISL and these antagonistic mutations are likely complex, and these hypotheses would need to be tested in future studies to confirm their validity.”

We have also included a reference that refers to the mechanism of action of K65R on ISL hypersusceptibility. Please refer to line 515 – 517: “The mechanism of hypersuceptibility of K65R to ISL has been shown to be mainly through decrease excision of ISL from the DNA terminus (Michailidis et al 2013).”

Comment 3:

K70T not hypersusceptible as stated on 366 Fold change not 1.5?

Response:

Please see comment 1.

Round 2

Reviewer 2 Report

Comments and Suggestions for Authors

The authors have responded adequately to my comments. I have only one further recommendation: to change the charts in Figures 2, 3, 4A, and 5 back to columns with error bars and overlaying data points. The symbols in the new versions are less easy to interpret than the originals.

Author Response

We thank the Reviewer for the recommendation. We have now changed the figures to bar graphs to represent the median values, with the IQRs represented as error bars.